# Mechanistic Studies for Palladium Catalyzed Copolymerization of Ethylene with Vinyl Ethers

**DOI:** 10.3390/polym12102401

**Published:** 2020-10-19

**Authors:** Andleeb Mehmood, Xiaowei Xu, Waseem Raza, Ki-Hyun Kim, Yi Luo

**Affiliations:** 1State Key Laboratory of Fine Chemicals, School of Chemical Engineering, Dalian University of Technology, Dalian 116024, China; andleeb.mehmood@gmail.com (A.M.); xuxiaowei001@mail.dlut.edu.cn (X.X.); razawaseem2@yahoo.com (W.R.); 2Department of Civil and Environmental Engineering, Hanyang University, 222 Wangsimni-Ro, Seoul 04763, Korea

**Keywords:** DFT, vinyl ethers, palladium phosphine sulfonate

## Abstract

The mechanism of ethylene with vinyl ether (VE, CH_2_=CHOEt) copolymerization catalyzed by phosphine-sulfonate palladium complex (**A**) was investigated by density functional theory (DFT) calculation. On achieving an agreement between theory and experiment, it is found that the favorable 1,2-selective insertion of VE into the complex **A** originates from stronger hydrogen interaction between the oxygen atom of VE and the ancillary ligand of catalyst **A**. Additionally, VE insertion is easier into the ethylene pre-inserted intermediate than that into the catalyst to form the resultant copolymers with the major units of OEt in chain and minor units of OEt at the chain end. The effect of *β*-OEt and *β*-H elimination was explored to elucidate chain termination and the molecular weight of copolymers. Furthermore, a family of cationic catalysts has been demonstrated to copolymerize ethylene with VE along with our modified cationic complex **B** with higher incorporation of VE and reactivity in comparison with complex **A**, which was modelled computationally by increasing the strong interactions between the catalyst and monomer moiety. Other than VE, the activity of cationic complex **B** for copolymerization of vinyl chloride and methacrylate is also computed successfully.

## 1. Introduction

Incorporation of polar functional groups is an effective option to improve polyolefins’ properties [1]. It is known that transition metal catalyzed copolymerization has become a powerful tool for synthesizing various copolymers with polar functionalized co-monomers [2,3,4,5,6,7]. However, copolymerization reactions of olefin and polar monomers is a challenge due to the low incorporation rates of polar monomers and the Lewis base poisoning effect towards positively-charged (Lewis acids) metal centers. Among polar vinyl monomers, the production of linear copolymers from vinyl ethers have been extensively investigated. These polar vinyl monomers are highly attractive among various synthetic copolymers (poly (vinyl ethers)) due to their tunable OR (O = oxygen and R = ethyl, butyl, tertiary butyl, and phenyl) groups [8,9]. In comparison to other alkenes, vinyl ethers have electron-rich π-bonds with enhanced reactivity. In 2006, it was reported that the branched (co)polymers poly(hexane co-CH_2_–CHOSiPh_3_) could be yielded by (α-diimine)PdMe+ catalyzed copolymerization of silyl vinyl ethers (CH_2_=CHOSiPh_3_) with 1-hexene [10]. Subsequently, these authors investigated the possible copolymerization mechanism for vinyl ethers (CH_2_=CHOR) with a different substituting OR group in (α-diimine)PdMe + catalytic system. In the cationic polymerization, fast insertion of a less electron-rich monomer CH_2_=CHOSiPh_3_ and relatively low *β*-OR elimination was observed. In contrast, for more electron-rich vinyl ethers, *β*-OR elimination was faster than insertion, suggesting that co-monomer of 1-hexene and more electron-rich vinyl ethers are not suitable in the cationic polymerization [11]. Furthermore, copolymerization of ethylene vinylalkoxysilanes (CH_2_=CHOSiPh_3_) was reported by using a traditional Pd(II) catalyst to explain their chain transfer mechanism and the *β*-silyl elimination [12]. In contrast to cationic α-diimine palladium complexes (α-diimine)PdMe+), a (phosphine-sulfonate) PdMe catalyst was proposed to copolymerize ethylene and alkyl vinyl ethers. This process generates linear alkyl copolymer units (vinyl ether in-chain and chain-end) with lower reactivity. It was found that incorporation of polar vinyl monomers increased significantly (12.5 mol%) during the copolymerization of ethylene and divinyl formal than that of vinyl ether (6.9 mol%) by the same catalyst [13,14,15]. In another study, a modified palladium catalyst bearing imidazole [1,5-a]quinolin-9-olate-1-ylidene (IzQO) was successfully used to suppress the *β*-hydride elimination during the copolymerization of the functionalized monomers [16]. Meanwhile, a polyethylene glycol unit was substituted on some phosphine sulfonate palladium and nickel catalysts. This strategy proved to be useful to acquire better catalytic activity, stability, and molecular weight of ethylene polymerization [17]. Moreover, cationic palladium complexes ligated by bisphosphine monoxide (BPMO) demonstrate higher reactivity of these catalysts as compared to the Brookhart-type complexes. In addition, bisphosphine monoxide (BPMO) also produces highly linear copolymer microstructures with a random distribution of polar functional groups throughout the polymer chain [18]. Further, investigations to improve the reactivity and incorporation by modifying the ligand platform, backbone, or substituting group has provided new routes for the catalyst tuning [19,20,21,22]. To date, the low incorporation of polar monomers and copolymers with low molecular weight is still a challenge. On this note, by following up catalyst **A** and the associated limitations of cationic palladium catalysts has stimulated us to explore more about cationic palladium systems for copolymerization of ethylene polar vinyl monomers [16,18,21,23]. Thus, we modified (PO)PdMe catalyst (**A**) computationally into catalyst **B** bearing a cationic bis-phosphine monoxide (BPMO) ligand, by replacing the sulfonate moiety with phosphine monoxide, which greatly influenced catalytic activity and the insertion mechanism of the VE monomer. To make this study more systematic we considered the experimentally introduced catalysts relative to the class of **B**.

In this manuscript, the chain initiation, propagation, and termination of ethylene and vinyl ether copolymerization in catalyst **A** have been investigated in detail. Further, we have theoretically explained the reason of low incorporation of vinyl ether, the activity of catalyst, and the low molecular weight of copolymer as reported previously [15]. The modified catalyst (**B**) is proposed as an effective route to improve the catalytic activity in copolymerization relative to catalyst **A**. We obtained the copolymer units as shown in Scheme 1 with better incorporation of vinyl ether (up to 69.9%) by using our modified catalyst **B**, which was quite low in the system (**A**) (up to 3.3%), which may help to improve intrinsic properties (flexibility and adhesion) of polymers. Including the effect of different substituents on the phosphorous, we also focused on the difference of activity shown by some fundamental polar monomers (methacrylate and vinyl chloride) by using **A** and **B**.

## 2. Computational Methods Used to Elucidate the Copolymerization

Density functional theory (DFT) calculations were used to investigate the mechanism of copolymerization of ethylene and VE. All the calculations were carried out using the Gaussian 16 program [24]. The B3PW91 functional [25] was selected for geometric optimization and frequency analysis of all structures. The effective core potentials (ECPs) of Hay and Wadt with a double-ζ valence basis set (LanL2DZ) was used for Pd atom [26,27,28], while the 6-31G* basis set was used for the rest of atoms; such basis sets are denoted as BSI. To obtain more reliable relative energies, single-point calculations of optimized structures were performed further at a higher level using the dispersion-corrected [29] density functional method B3PW91-D3 together with BSII. In BSII, the Stuttgart/Dresden ECP and associated basis sets were applied to the Pd atom, while 6-311G(d,p) was used for the rest of the atoms [30,31,32,33]. Toluene was used as a solvent to match the experimental data. Thus, the solvation effect of toluene was considered through the SMD model in these single-point calculations [34]. The energy profiles of the insertion mechanism were described by the relative free energy in solution phase (ΔG, kcal/mol) [35]. Optimized geometrical structures of transition states were illustrated using CYLview [36]. Non-covalent interaction analysis (NCI) was performed for some transition states (TSs) using Multiwfn and VMD software to observe the weak interactions between catalysts and monomers [37,38]. Cavallo’s SambVca 2.0 program was also used to visualize the steric hindrance of ligands around a central metal [39].

## 3. Results and Discussion

### 3.1. Chain Initiation

The dissociation of neutral ligand (Py) easily occurs from complex **A** and yielding complex **A′** could be a true initial active species. For the electronically asymmetric features of phosphine sulfonate-type catalysts, the monomer could coordinate with metal centers from two sites, viz., *cis* and *trans* (for the *cis* site, the monomer coordination is on the opposite side of the P atom of the phosphine group, while for the *trans* site, monomer coordination is on the opposite side of the O atom of the sulfonate group) (see Appendix A). It is acknowledged that the process firstly goes through the *trans* complex and is isomerized to the less stable *cis* complex, and then adopts the preferable *cis*-insertion kinetically in the system of polymerization reaction by phosphine-sulfonate catalysis [40,41,42,43,44]. Based on this, the mechanism of VE insertions have been explored in detail with different modes (viz., 1,2 and 2,1-modes).

Note that the coordination complex (**1A__12VE_**) of the double bond of VE with **A′** is more stable than that of oxygen atoms of VE (**1A__12VEO_**) (see Appendix A). As Figure 1 shows, 1,2-insertion of VE started form the π-coordination complex (**1_A_12VE_**) with *trans* fashion firstly releasing much energy (15.1 kcal/mol). After that, it needs to transform into the complex (**2_A_12VE_**) with *cis* fashion. In order to proceed through the favorable pathway, *cis*-insertion takes place via a transition state (TS), **3TS_A_12VE,_** and it yields the most stable product (**4_A_12VE_**) with O-chelated interaction, whereas the *cis* and *trans* complexes of the VE 2,1-coordination mode are slightly more stable than those of 1,2-coordination mode, respectively. In addition, the **3TS_A_21VE_** with 2,1-insertion mode is less stable than **3TS_A_12VE_** with 1,2-insertion mode, and the free energy barrier of 2,1-insertion is much higher than that of 1,2-insertion (27.5 vs. 25.6 kcal/mol). Therefore, 1,2-insertion of VE is more favorable in comparison with 2,1-insertion in the aspects of kinetics and thermodynamics, which is in line with experimental results [15]. To explore the reason why 1,2-insertion is favorable, a comparative distortion/interaction analysis [45,46,47,48] was performed for TSs, **3TS_A_12VE_**, and **3TS_A_21VE_**. They are divided into two fragments: monomer moiety (fragment B) and the remained catalyst moiety (fragment A). In TS, geometrical energies of fragments A and B were evaluated by single point calculations. The interaction energies (**∆*E***_int_) of fragments A and B were calculated by estimating single point energies along with the energy of TS. To calculate **∆*E***_TS,_ total deformation energy (∆*E*_dist_) was calculated first by combining the deformation energies of each fragment, i.e., ∆*E*_dist(A)_ and ∆*E*_dist(B)._ Finally, **∆*E***_TS_ was estimated using the relation ∆*E*_TS_ = ∆*E*_int_ + ∆*E*_dist(A)_ + ∆*E*_dist(B)_ as shown in Figure 2. Although the total distortion energy of fragment A and B in the **3TS_A_12VE_** is almost same as **3TS_A_21VE_** (39.9 + 21.7 = 61.6 vs. 39.8 + 21.7 = 61.5 kcal/mol), the interaction between the two fragment in **3TS_A_12VE_** is much more negative than that in **3TS_A_21VE_** (−60.3 vs. −58.7 kcal/mol). Therefore, the ∆*E*_TS_ (61.5 − 60.3 = 1.2) obtained for 3**TS_A_12VE_** is lower than that for **3TS_A_21VE_** (61.6 − 58.7 = 2.9). Therefore, the higher stability of 3**TS_A_12VE_** is mainly due to the stronger interaction between the catalyst and monomer moiety. This is indicated by the short distance of Pd•••C2 and extra hydrogen interaction between O4•••H1 in **3TS_A_21VE_** in comparison with **3TS_A_21VE_**
Figure 2. Additionally, steric maps are also showing more hindrance for **3TS_A_21VE_** (81.4 vs. 80.4%V_Bur in **3TS_A_12VE_**) (see Appendix A).

The computed energy profile of ethylene insertion into Pd-C of active species (**A**) is presented in Figure 1. The formation of π-coordination complex (**1_A_E_**) in *trans* fashion and complex (**2_A_E_**) with *cis* is less stable than those of VE, respectively. This can be attributed to the lack of hydrogen interaction in **1_A_E_** as seen in Appendix A. After **1_A_E_** the ethylene will overcome the free energy barrier of the 20.0 (8.9 + 11.2) kcal/mol. In this process the product (**4_A_E_**) with *γ-*agostic interaction could be yielded to be isomerized readily to more stable product (**4′_A_E_**) with *β*-agostic interaction after releasing the energy of 2.5 (18.6 − 16.1) kcal/mol. It is obvious that the free energy barrier of ethylene insertion is 20 kcal/mol, which is much lower than that of VE (25.6 kcal/mol, Figure 1). To have a better understanding on the activity difference in insertion between ethylene and VE, a similar distortion/interaction analysis has been performed for the **3TS_A_E_** (Figure 2c). The total distortion energy (∆*E*_dist_ 56.1 (16.3 + 39.8) kcal/mol) can be balanced by its interaction energy (∆*E*_int_ −57.8 kcal/mol), which is leading ∆*E*_TS_ by −1.7 kcal/mol. Note that the interaction between two fragments in the **3TS_A_E_** is less negative than that of **3TS_A_12VE_** (−57.8 vs. −60.3 kcal/mol). Thus, the smaller distortion of ethylene is a main factor of higher stability for **3TS_A_E_** than that of VE. In Appendix A. geometric analysis has also shown that the total difference in bond angles for ∠C1‒C2‒H2 of 1,2-insertion of VE is larger (∆(∠C1‒C2‒H2) = 3.7°) than the ethylene (∆(∠C1‒C2‒H2) = 3.0°). On the other hand, VE contains an OR group with a very large difference of angle (∆(∠C1‒C2‒O4)= 7.0°) than the hydrogen of ethylene (∆(∠C1‒C2‒H4) = 3.8°) due to the presence of a large OR group in VE. It is already known that large size atoms undergo more distortion due to more polarizability. Hence, the presence of an oxygen atom in VE causes more distortion than the small size of hydrogen of ethylene.

It is obvious that the coordination complex of VE should be much more stable than that of ethylene, as observed by the less negative coordination energy of the former (−28.2 kcal/mol) than that of ethylene (−24.0 kcal/mol). In contrast, the insertion of ethylene is favorable than that of VE, as the activation barrier of the former is lower than the latter by 2.5 kcal/mol. By taking account of the kinetic and thermodynamic aspects, we estimated the probability ratio of VE insertion into the initial active species (**A′**) at the chain initiation stage. The population ratio between the complex coordinated with VE and the ethylene, n_E_/n_12VE_, can be calculated in accordance with Boltzmann statistics [49,50]:(1)$$\frac{n_{E}}{n_{12VE}}=exp\left(-\frac{∆G_{C,E}}{RT}\right)/exp\left(-\frac{∆G_{C,12VE}}{RT}\right)$$

Here ΔG_C,E_ and ΔG_C,12VE_ denote the coordination free energies for the insertion of ethylene and 1,2-VE insertion, respectively. Note that n_E_ and n_12VE_ represent the population of ethylene-coordinated complex (**1_A_E_**) and VE 1,2-coordinated complex (**1A__12VE_**), respectively (R = 8.3145 J·mol^−1^·K^−1^ and T = 298.15 K). At the stage of chain initiation, the values for the population of ethylene-coordinated complexes were computed as: Δ*G*_C,E_ = −11.2 kcal/mol; Δ*G*_C,12VE_ = −14.8 kcal/mol; and n_E_/n_12VE_ = 0.0023. Moreover, on the basis of this population ratio and the insertion free energy barrier, the probability ratio of the ethylene-insertion and 1,2-insertion into the active species, P_E_/P_12VE_, can also be estimated according to the equation:(2)$$\frac{P_{E}}{P_{12VE}}=\frac{n_{E}}{n_{12VE}}exp\left(-\frac{∆G_{E}^{‡}}{RT}\right)/exp\left(-\frac{∆G_{12VE}^{‡}}{RT}\right)$$

Δ*G*_E_**^‡^** and Δ*G*_12VE_**^‡^** denote the insertion free energy barriers for ethylene-insertion and VE 1,2-insertion. At the stage of chain initiation, the probability ratio of the ethylene vinyl ether 1,2-insertion into the initial active species (**A′**)2 was computed as: Δ*G*_E_**^‡^** = 20.0 kcal/mol; Δ*G*_12VE_**^‡^** = 25.6 kcal/mol; P_E_/P_12VE_ = 29.23. It is suggested that the 1,2-insertion of VE shows approx. 3.3% probability while that of ethylene shows approx. 96.7% probability. Therefore, ethylene gained an overwhelming advantage over VE for the insertion during the chain initiation stage.

### 3.2. Chain Propagation

Subsequently, the mechanism of ethylene and VE insertion into the ethylene pre-inserted intermediate (**4′_A_E_**) and VE pre-inserted intermediate (**4_A_12VE_**) were explored during the stage of chain propagation. The computed energy profiles of ethylene and VE insertion with optimal fashion into **4′_A_E_** were presented in Figure 3. Similar to the case of chain initiation in Figure 1, the formation of coordinative complex **5_A_E_** of ethylene with **4′_A_E_** is less stable than that of **5_A_12VE_** in *trans* fashion. Successive ethylene insertion proceeds through **4′_A_E_** → **5_A_E_**→ **6_A_E_**→ **7TS_A_E_** → **8_A_E_**, the free energy barrier of this process is 20.6 (24 − 3.4) kcal/mol and is exergonic by 11.5 (30.1 − 18.6) kcal/mol. Yet, VE insertion into **4′_A_E_** proceeds through **4′_A_E_** →**5_A_12VE_**→**6_A_12VE_** →**7TS_A_12VE_** →**8_A_12VE,_** and the free energy barrier of this process is 23.1 (28.2 − 5.1) kcal/mol, which is lower than the insertion into **A** at the chain initiation stage (25.6 kcal/mol, Figure 1). This is due to the extra hydrogen interaction between the longer growing chain and incoming VE monomer Appendix A.

Next, the third monomer insertion were calculated into ethylene and the VE pre-inserted intermediate (**8_A_12VE_**). The activation barrier of 28.7 kcal/mol for VE insertion is much higher 5.9 (28.7 − 22.8) kcal/mol than that of ethylene insertion, which is attributed to the rather stable π-coordination complex (**9_A_12VE_**) and less stable TS (**11TS_A_12VE_**) in comparison with the corresponding stationary points, respectively. Further calculations may be performed to clarify the discrepancy in the two monomer insertion. The bonding energy Δ*E* (bonding) of **9_A_12VE_** is −29.5 kcal/mol, which is more negative than that of **9_A_E_** (−25.3 kcal/mol), suggesting the stronger interaction in the former. This is indicated by the hydrogen interaction between the growing chain and VE (see Appendix A). Meanwhile, the less steric hindrance could lead to a lower activation barrier, which is indicated by a comparison between the steric maps of **11TS_A_12VE_** and **11TS_A_E_**, in which the latter has much less hindrance in the SW quadrant Figure 4a,b.

It has been found that once VE insertion occurs, the next insertion of ethylene will become harder and lead to lower reactivity and decreasing molecular weight. This is suggested by the harder coordination and insertion in comparison with ethylene insertion into **4_A_E_** (0.8 vs. −5.4 for ethylene coordination, 23.2 vs. 20.6 kcal/mol for ethylene insertion). Meanwhile, ethylene insertion still holds on to a favorable position in comparison with VE insertion after last VE insertion, which is consistent with the experimental result that the copolymer units with OEt in chain could be obtained and there are few VE continuous insertion units [15].

At the initiation stage, the insertion of VE adopts the 1,2-fashion predominately according to the energetic factors. The insertion of ethylene and VE into the VE pre-inserted intermediate (**4_A_12VE_**) was investigated during the chain propagation. As shown in Figure 5, weak interaction of the monomer and catalyst is responsible for the slightly endergonic coordination processes of ethylene in **13_A_E_**, as compared to the exergonic VE in **13_A_12VE_** (see Appendix A). During the insertion process of ethylene, the relative free energy of **15TS_A_E_** (21.2 kcal/mol) is slightly higher than that of **3TS_A_E_** (20.6 kcal/mol) by 0.6 kcal/mol. It is indicated that the insertion of ethylene into VE pre-inserted active species is kinetically less favorable than the continuous insertion of ethylene. A successive insertion of VE is difficult due to the high energy barrier of 29.1 kcal/mol. These calculated results are in line with the observed yield of minor copolymer units with OEt at the chain end and without VE continued units.

To sum up, the VE could be inserted into the ethylene pre-inserted intermediate with an overall free energy barrier of 23.1 kcal/mol through the pathway A of **1_A_E_**→**3TS_A_E_**→**4′_A_E_**→**5_A_12VE_**→**7TS_A_12VE_**→**8_A_12VE_** to form the growing units with OEt in the chain. In contrast, ethylene could be inserted into the VE pre-inserted intermediate with the overall free energy barrier of 25.6 kcal/mol through the pathway B of **1_A_12VE_**→**3TS_A_12VE_**→**4_A_12E_**→**5_A_E_**→**7TS_A_E_**→**8_A_E_** to form the growing units with OEt at the chain end. Thus, for the growing unit, the presence of a lower free energy barrier for pathway A ultimately suggests that the OEt content within the chain is more than that at the chain end. Additionally, the free energy barrier of the continuous insertion of ethylene is much lower (20.6 kcal/mol) than that of VE insertion (pathway A 25.6 vs. pathway B 23.1 kcal/mol). Therefore, the copolymers feature mainly alkyl chains and few of VE units in the chain or chain end. This is in agreement with experimental observation that the NMR detected with low incorporation ratios of VE.

### 3.3. β-H and β-OEt Elimination

As for the observed decreasing reactivity and low molecular weights of copolymerization in the experiment [15], the chain termination needs to be discussed and its mechanism seeks to be clarified. Therefore, the computed energy profiles of *β*-OEt and *β*-H elimination are followed by chain re-growth, as presented in Figure 6. The *β*-OEt elimination process started from the intermediate (**4_A_12VE_**) formed by 1,2-VE insertion of VE and goes through the pathway **4_A_12VE_**→**5*_β_*__OEt_**→**6TS*_β_*__OEt_** →**7*_β_*__OEt_** (green line in Figure 6). This process needs to overcome the free energy barrier of 28.3 (6.9 + 21.4) kcal/mol with the endergonic species **7*_β_*__OEt_** by 9.4 (21.4 − 12) kcal/mol, thus suggesting the *β*-OEt elimination process is kinetically and thermodynamically unfavorable. The Pd-OEt active species hardly exists in this system, which is a different form of the α-diamine palladium system [51]. Further, ethylene insertions into the Pd−OEt bond of **7*_β_*__OEt_** suffer from a very high energy barrier (27.5 kcal/mol), which is higher by 6.3 kcal/mol (=27.5 − 21.2 kcal/mol) for chain propagation, as shown in Figure 6. Additionally, the *β*-H elimination pathway was explored based on the intermediate (**4′_A_E_**), which was formed by ethylene insertion and isomerization of species **4_A_E_** (black line in Figure 6). As seen in Figure 6, **4****′_A_E_** needs to overcome a lower free energy barrier of 9.4 (18.6 − 9.2) kcal/mol (Intrinsic reaction coordinate (IRC), Appendix A) which is slightly endogenic by 1.8 (18.6 − 16.8) kcal/mol. It is thus suggested that the *β*-H elimination may result in a quick-reversible process during the high-pressure insertion reaction of ethylene in terms of kinetics and thermodynamics. Herein, a facile release of propene occurs, and Pd-H species coordinate with ethylene as **12_E_**. Further, direct insertion of an incoming ethylene molecule (**14TS_E_**) allows the *β*-agostic complex **15_E_** to achieve its chain transfer. The overall free energy barrier of chain transfer is 24.3 (5.7 + 18.6) kcal/mol, which is higher than that of chain propagation (20.6 kcal/mol, Figure 3). This phenomena indicates that the chain transfer could occur to a certain extent to decrease the molecular weight and reactivity, which is consistent with the experimental observation [15].

### 3.4. Catalyst Tuning

As for system **A**, there are a number of limitations of low incorporation ratios, catalytic activity, and molecular weight in the copolymerization of ethylene and vinyl ether. With these insights, we attempted to estimate the polymerization behavior of ethylene-VE catalyzed by cationic catalysts (BPMO) demonstrated in previous studies, which has shown some unique features in the polymerization of ethylene and polar monomers in comparison with the neutral catalyst **A** [18,21,23]. Firstly, cationic palladium complexes ligated by a bisphosphine monoxide (BPMO) were observed to be suitable for ethylene polymerization. We also compared the experimental results with our computational data. Accordingly, high energy barriers of catalyst **b** and **c** (20.4 and 19.0 kcal/mol) were observed as compared to **a** (18.0 kcal/mol) for ethylene polymerization. This observation thus suggests that polymerization should become easier when the substituent of phosphine is isopropyl rather than phenyl or *o*-methoxy phenyl group. Higher activity of **a** is attributed to the smaller distortion of catalyst and monomer as compared to the **b** (11.7 + 36.4 = 48.1 vs. 13.0 + 38.4 = 51.4 kcal/mol) (Appendix A) Table 1. From **d** to **e,** substituents on phosphine are kept the same while changes are performed at the phosphine oxide group by the replacement of tertiary butyl to isopropyl groups. Herein, an increase in the energy barrier is seen as 18.4, 18.7, and 19.0 kcal/mol for **d**, **e**, and **c**, respectively. These findings are consistent with the experimental results in that the activity of ethylene polymerization is higher for **a** catalyst with the lowest energy barrier of 18.0 kcal/mol than that of **b** (20.4 kcal/mol) [12].

For catalysts **a**, **b**, **c**, and **d**, almost similar energy barriers (15.0, 15.6, 15.3, and 16.2 kcal/mol) of copolymerization are observed. It can thus be concluded that the change in the substituent on phosphine should have negligible effect on copolymerization reactivity although it may have partial effects on the ethylene polymerization. As the energy barrier of catalyst **d** is lower (16.2 kcal/mol) than that of **e**, the change in the substituent of phosphine oxide from tertiary butyl (in catalyst **d**) to isopropyl (in catalyst **e**) may be more effective for copolymer formation.

Large differences in the energy barriers of **d** and **e** are further evaluated to understand the substituent effect on catalytic activity during copolymer formation. From distortion/interaction analysis, we found that the combination of stronger interaction energy of **d** as compared to **e** (−70.4 vs. −69.6 kcal/mol) and less distortion (13.5 + 22.7 = 36.2 vs. 23.0 + 16.0 = 39 kcal/mol) contribute to the stability of the transition state during copolymer formation, as shown in Figure 7. It is worth noting that a small isopropyl group is more favorable to rotate in the available space around the phosphine oxide of catalyst **e** as compared to the large tertiary butyl group of catalyst **d**. The catalyst fragment of transition state E-VE^TS3^ in **d** system showed larger changes of dihedral angles among ∆(C6‒P‒O‒Pd) than **e** (3.05° vs. 6.9° in **e**), likewise for ∆(C5‒C6‒P‒O) (2.86° vs. 31.0° in **e**). A previous study also confirmed that the large ring size increases the catalytic flexibility to ultimately affect the electronic parameters of the metal center. Small ring sizes in Pd complexes cause the rigidity in the catalyst with the enhancement of the catalytic properties [52].

On the basis of our computational findings obtained from the study of cationic catalysts (BPMO), further we modified catalyst **B** computationally by the replacement of ‒SO_2_‒O^−^ (in catalyst **A**) with P(Me_2_)=O^+^. As followed by the **A** system, the two pathways of forming the copolymer of ethylene and VE insertion were located as shown in Figure 8.

For the chain initiation step, the ethylene coordination is less stable than VE, as suggested by the free energies between the former (−10.7 kcal/mol) and the latter (−13.8 kcal/mol). Immediately after the coordination, the ethylene is easier to insert than that of VE, which is manifested by their free energy barriers of 17.7 and 20.8 kcal/mol, respectively. After that, the probability of VE insertion was estimated through the Boltzmann equation (vide supra) by considering the coordination and insertion in total. The 1,2-insertion of VE showed the probability of approx. 69.9%, which is much higher than that of system **A** (3.3% probability). This indicates that the incorporation ratios of VE may have been improved in system **B** relative to **A**. Meanwhile, it was found that the insertion of ethylene and VE into **B′** is much easier than into **A′**.

It is noted that the free energy barrier of ethylene insertion into **B****′** (2.9 (20.6 − 17.7) kcal/mol) is lower than that of **A′**. We also observed that the free energy barrier of VE insertion into **B′** is approximately 5.4 kcal/mol, which is far lower than that of **A′** (25.4 kcal/mol) (see Figure 8). This can be attributed to the stronger interaction between the catalyst **B′** and monomer moiety (−58.1 kcal/mol for ethylene, −65.7 for VE) as compared to **A****′** (−57.8 for ethylene, −60.3 kcal/mol for VE). The shorter bond lengths of Pd•••C1 and Pd•••C2 are seen in **3TS_B_E_** and **3TS_B_12VE_** than in **3TS_A_E_** and **3TS_A_12VE_**, respectively (see Figure 2 and Figure 9, respectively). Moreover, a natural bond orbital analysis of the same species between **3TS_B_E_** and **3TS_B_12VE_** indicates some more interactions as there are shorter bond lengths for H5•••O5 (2.71 vs 3.51 and 2.77 vs. 3.53, respectively) (see Appendix A). Furthermore, in Appendix A, the LUMO energy of **B′** (as compared to **A′**) is much closer to the HOMO energy of monomers E and VE, which ultimately confirms the applicability of catalyst **B** as compared to **A**.

In addition to VE, we also examined the advantage of catalyst **B** for copolymerization of ethylene with other fundamental polar vinyl monomers, like methacrylate (MA) and halogenated vinyl monomer as vinyl chloride (VC), which are inexpensive and readily available. Methacrylate and vinyl chloride display a 2,1-insertion mode as compared to VE (1,2-insertion) in both systems **A** and **B**. To observe the activity of **B**, we considered only the copolymer formation step. Hence, VC and MA monomers are inserted in to the ethylene pre-inserted intermediates (**4_A_E_** and **4_B_E_**). It is noted that the free energy barriers of VC and MA with catalyst **B** are quite lower (17.6 and 14.9 kcal/mol) than that of **A** (19.1 and 17.2 kcal/mol, respectively). Calculated low energy barriers by using **B** suggested that changes in the ligand of catalysts can the design of high-performance catalysts for challenging polar vinyl monomers.

## 4. Conclusions

The copolymerization mechanism of ethylene and VE catalyzed by Pd catalyst (**A**) has been explored by DFT calculation. It has been found that VE insertion can be favorable with the 1,2-modes in comparison with 2,1-modes, which is due to the stronger interaction between the catalyst and monomer in the former mode. In contrast, ethylene is harder to coordinate with Pd-Me than that of VE. This can be attributed to the existence of hydrogen interactions between the oxygen atom of VE and the catalyst ligands. After insertion of ethylene, the subsequent insertion of VE needs to overcome a higher free energy barrier of 23.1 kcal/mol than that of continuous insertion of ethylene (20.6 kcal/mol). This indicates that the catalytic activity of copolymerization of ethylene and VE is lower than that of the homopolymerization of ethylene. Once VE is inserted in the Pd-Me specie, the ethylene insertion becomes harder than the continuous insertion of ethylene. Nevertheless, repeated insertion of VE needs to overcome a quite high free energy barrier (29.1 kcal/mol) that ultimately makes this process kinetically unfavorable. This phenomena can be mainly attributed to steric hindrance between the coming VE and growing chain. The computational results are in good agreement with the experimental data in that the resultant copolymer only contains OEt in the chain and chain end without the repetitive insertion units of VE. Furthermore, as the *β*-OEt hardly occurs, the insertion of ethylene becomes more difficult. Moreover, after an easy *β*-H elimination and difficult reinsertion of ethylene in Pd−H there may be a decrease in the molecular weight and reactivity as well.

In addition, we considered the cationic family of Pd catalyst (BPMO) to evaluate the effect of different substituted ligands. We found that tertiary butyl substituted at phosphine oxide is effective for ethylene and VE copolymerization with the least distortion of catalyst **d**. Our modified cationic complex **B** was calculated and found more active in improving the reactivity and incorporation ratios of polar monomer in the system of ethylene with VE copolymerization. A stronger interaction has been observed between catalyst and monomer in system **B** relative to the neutral catalyst **A**. Finally, other fundamental polar monomers (VC and MA) have also shown better activity for the copolymerization step in system **B**. This study may further provide better understanding of the mechanism of copolymerization of ethylene and polar vinyl monomers when catalyzed by Pd metal complexes.

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
