# Peer review of "Mechanistic Studies for Palladium Catalyzed Copolymerization of Ethylene with Vinyl Ethers"

_polymers, 2020, doi:10.3390/polym12102401_

Round 1

Reviewer 1 Report

This manuscript from Luo, Kim and coworkers investigated the copolymerization between ethylene and vinyl ether through computational modeling. The thermodynamics of each step are calculated, which provide the community with unique insights about this polymerization. The authors also designed a new set of Pd catalysts with various ligands. A preliminary structure-reactivity relationship of those cationic Pd species has been discussed. Overall, this computational study not only fill out the missing pieces of current data, but also provide valuable potential catalysts for future experiments. I suggest to accept this manuscript in current form.

Reviewer 2 Report

The authors have addressed all of my concerns. The manuscript has been improved after the revision. I might suggest its acceptance. 

This manuscript is a resubmission of an earlier submission. The following is a list of the peer review reports and author responses from that submission.

Round 1

Reviewer 1 Report

Manuscript Title:

Mechanistic studies for palladium catalyzed copolymerization of ethylene with vinyl ethers

Manuscript ID : polymers-929911

Recommendation: Publish in Polymers after minor revisions

Comments:

This manuscript provides a comprehensive computational study to investigate the mechanisms of Pd-catalyzed copolymerization of ethylene with vinyl ethers. The processes of chain initiation, propagation, and termination have been studied in this manuscript. The authors also investigated the origin of reactivity and selectivity using interaction/distortion model, and steric maps. I believe this paper would be interesting to both experimental and computational chemists in the field of polymer chemistry and organic chemistry. However, a few concerns need to be addressed before further consideration of publication in Polymers.

  1. The chemical formulas for vinyl ethers should be modified in the manuscript

The double-bond in vinyl ethers should be written explicitly in the chemical formulas. For example, in line 40, silyl vinyl ethers should be "CH2=CHOSiPh3" instead of "CH2-CHOSiPh3"; in line 41, vinyl ethers should be " CH2=CHOR" instead of " CH2-CHOR".

The authors should go through the full manuscript carefully to correct those.

  1. Issues with Figure 1

1) Figure 1 has to be modified since there are two 2A_E 2D structures (in blue) in the energy profile. I think the authors should put 2A_12VE structure instead.

2) Why the barriers of VE insertion transition states (3TSA_VE) are higher than ethylene insertion transition states (3TSA_E) while the intermediates before and after VE insertion (2A_VE & 4A_VE) are more stable? Is there any difference in optimized geometry between transition states and intermediates?

  1. Questions about the origin of reactivity & 1,2 selectivity for VE insertion

1) In Figure 2, the distortion/interaction analysis has been applied, and the 1,2-VE insertion has stronger interaction energies. The authors claimed that it was due to short distance of Pd-C1 and hydrogen interaction between O4-H1. I would not buy this explanation. I think the author should consider the unfavorable repulsion between OR group and Ar group in the catalyst in 2,1-VE insertion transition state, which may lead to the longer Pd-C1 distance in 3TSA_21VE.

I will suggest the author should apply steric maps or percent buried volume calculations to demonstrate the steric effect on 1,2 selectivity.

2) In Figure 2, it seems that the VE in the transition states has been much more distorted than the ethylene with > 5 kcal/mol distortion energy difference. What causes this huge difference? More computational analysis or discussion are needed.

  1. The transition state symbol should be double-dagger (‡) instead of unequal symbol (≠). The authors should change all of them in the figures and in the manuscripts.

Reviewer 2 Report

This manuscript from Mehmood et al. applied DFT calculation to investigate the mechanism of palladium catalyzed copolymerization between ethylene and vinyl ethers. Although both computation itself and the way results presented are sound and decent, this manuscript lacks publication-level novelty to the polymer community, especially for those working on coordination-insertion polymerization. The research described just seems to be some trivial computational addition to these two references authors cited in the paper:1) Luo et al., J. Am. Chem. Soc., 2007, 129, 8946-8947. 2) Carrow et al., J. Am. Chem. Soc., 2012, 134, 8802-8805. Except for the catalytic performance of two well-known catalyst, there is no further knowledge that the community can learn from this paper. As a fully computational paper, this manuscript should provide visionary insight on at least one kind of novel Pd catalyst. So, I think this manuscript should be "rejection and resubmission".

To make this study more systematic, the authors should consider DFT calculation on a whole family of Pd catalyst B with different substituents on the ligand, focusing on the steric and electronic effect. Or, instead of looking at virtual catalyst which lacks real-world data, it would be nice if authors could do computation on those Pd catalysts used by Carrow et al. to test the accuracy between theory and experiment. Besides vinyl ether, the authors should look at a broader scope of polar vinyl monomers, which will certainly be more attractive to synthetic polymer chemists in the field.